# Sheep’s Second Cheese Whey Edible Coatings with Oregano and Clary Sage Essential Oils Used as Sustainable Packaging Material in Cheese

**DOI:** 10.3390/foods13050674

**Published:** 2024-02-23

**Authors:** Arona Pires, Hubert Pietruszka, Agata Bożek, Katarzyna Szkolnicka, David Gomes, Olga Díaz, Angel Cobos, Carlos Pereira

**Affiliations:** 1School of Agriculture, Bencanta, Polytechnic University of Coimbra, 3045-601 Coimbra, Portugal; 2Departamento de Química Analítica, Nutrición y Bromatología, Facultad de Ciencias, Campus Terra, Universidade de Santiago de Compostela, 27002 Lugo, Spain; olga.diaz.rubio@usc.es (O.D.); angel.cobos@usc.es (A.C.); 3Department of Toxicology, Dairy Technology and Food Storage, West Pomeranian University of Technology, Papieża Pawła VI St. No. 3, 71-459 Szczecin, Poland; 4Centro de Estudos dos Recursos Naturais, Ambiente e Sociedade—CERNAS, 3045-601 Coimbra, Portugal

**Keywords:** edible coatings, cheese, second cheese whey, essential oils, oregano, salvia

## Abstract

Sheep’s second cheese whey (SCW), the by-product resulting from whey cheese production, was used as a component of cheese coatings containing oregano (*Origanum compactum*) and clary sage (*Salvia sclarea*) essential oils (EOs). SCW powder was obtained by the ultrafiltration/diafiltration of SCW followed by reverse osmosis and freeze drying. The coatings were produced with a mixture of SCW and whey protein isolate (WPI) using glycerol as plasticizer. Model cheeses were produced with cow´s milk and those containing SCW:WPI coatings; those with and without EOs were compared to controls without coating and with a commercial coating containing natamycin. At the end of ripening (28 days), the cheeses containing EOs presented higher water activity (*ca.* 0.930) and moisture content, as well as lower titratable acidity. Concerning color parameters, significant differences were also observed between products and as a result of ripening time. However, the use of SCW:WPI coatings did not significantly influence the color parameters at the end of ripening. Regarding texture parameters, the cheeses containing SCW:WPI coatings presented significantly lower values for hardness, chewiness, and gumminess. Significant differences were also observed for all microbial groups evaluated either between products and as a result of ripening time. In all cases, lactobacilli and lactococci counts surpassed log 7–8 CFU/g, while the counts of yeasts and molds increased steadily from ca. log 3 to log 6 CFU/g. The lowest counts of yeasts and molds were observed in the samples containing natamycin, but nonsignificant differences between products were observed. In conclusion, SCW:WPI cheese coatings can successfully substitute commercial coatings with the advantage of being edible packaging materials manufactured with by-products.

## 1. Introduction

Whey cheeses (e.g., *Requeijão,* which is produced in Portugal) are obtained by heating cheese whey (CW) (90–95 °C for ca. 20 min) to precipitate whey proteins. The protein aggregates originate a matrix that entraps fat and other solids present in whey [1,2]. Second cheese whey (SCW) is the liquid remaining after whey cheese production and represents more than 90% of the original CW volume. Lactose (4.5–5.0%), salts (0.5–1.5%), proteins (0.15–0.3%), and fat (0.1–0.3%) represent the major solid components of bovine SCW [3,4]. SCW derived from sheep whey is typically characterized by a higher protein content, thus containing 7.5–10.5% dry matter, 4.2–5.5% lactose, 0.5–1.8% protein, 0.3–2.5% fat, and 0.5–1.0% minerals [4,5,6].

The high volumes of SCW produced (about 18 L per kg of whey cheese) represent a problem for cheese producers, as this by-product poses environmental challenges due to its very high chemical and biochemical oxygen demand (COD and BOD_5_ values higher than 30 g/L) [2,3,5]. Most SCW is used as supplement feed for livestock, either directly or as an additive [7], but high amounts are still discarded without treatment. However, this by-product has important nutrients such as denatured proteins, nonprotein nitrogen such as soluble peptides and free amino acids, oligosaccharides, lactose, and hydrosoluble vitamins and minerals [3,6,8,9].

SCW can serve as a substrate in biotechnological processes for producing high-value commercial compounds such as lactic acid, bioethanol, lactobionic acid or polyhydroxyalkanoates [5,10,11,12,13,14,15]. The obtention of bioactive peptides from SCW was also proposed [16,17]. Some manuscripts are related to the use of SCW as a biotechnological substrate for probiotic or starter cultures [18,19,20]. Others refer to its potential to produce biofuel products [21,22,23]. The use of SCW to produce fermented drinks or as an ingredient in food formulations, was also evaluated [3,24,25,26,27,28]. However, further research work on the valorization of SCW is still required, and the difficulties related with the technology transfer from researchers to dairy producers need to be overcome.

The utilization of dairy by-products on the production of edible films and coatings was generally based on the use of cheese whey (CW) [29,30,31,32]. Revisions on this subject have also been published [33,34]. CW films or coatings are often added with antimicrobial agents, namely essential oils obtained from plants such as cumin [35], clove [36], lemon, bergamot [37], oregano, rosemary, garlic [38,39], tarragon [40], and cinnamon [41]. Other plant-based ingredients were also tested [42,43,44,45]. Chitosan [46], lactic and propionic acids, chitooligosaccharides [47], and enterocins [48] were also evaluated in whey-based films. Several authors used previously fermented CW to produce edible films as carriers of lactic acid bacteria [49,50,51,52,53,54,55] or probiotics [56]. Finally, other research works refer to the proteins and peptides with biological activity present in CW films [57]. However, the evaluation of SCW in the production of edible films and coatings was almost ignored, and its potential as a sustainable packaging material was neglected. To the best of our knowledge, only one work reported [58] the use of SCW for such applications. The application of coatings based on whey proteins for cheeses is also limited. Whey protein isolates [59,60] and whey protein concentrates [61,62,63] have been used to elaborate coatings to wrap cheeses by incorporating antimicrobial compounds [59,60], natural plant extracts (cinnamon) [61], or protective lactic acid bacteria [62,63] to extend the shelf life of the product.

The aim of the present work was to test the efficacy of sheep’s SCW-based edible coatings on bovine milk cheeses. The cheeses containing SCW edible coatings, without the addition of antimicrobial agents (WC), with added oregano (*Origanum vulgare*) essential oil (WCO), or with clary sage (*Salvia sclarea)* essential oil (WCS), were compared with cheeses without coating (C) or with a commercial coating containing natamycin (N).

## 2. Materials and Methods

### 2.1. Production of Sheep Second Cheese Whey Powders

The sheep’s SCW was supplied by an external dairy company. A total of 500 L of sheep’s SCW were ultrafiltrated in a pilot plant (Proquiga Biotech SA, A Coruña, Spain) equipped with an organic ultrafiltration membrane (3838 PVDF/polysulfone) (effective filtration area of 7 m^2^ and 10 kDa cutoff) (FipoBiotech, Vigo, Spain). The process was performed at a temperature ranging between 40 and 45 °C, at a transmembrane pressure of 3–4 bar, with a volumetric concentration factor (VCF) of 20 (VCF = Vol. feed/Vol. retentate), thereby obtaining 25 L of concentrate. To the concentrate, 225 L of process water were added, and the diluted concentrate was submitted to diafiltration (VCF = 10). The diafiltrated concentrate was pasteurized (75 °C, 30 s) and was further concentrated by reverse osmosis (RO) in a pilot plant developed by ORM (Belas, Portugal) equipped with 2.5 S seawater pressure vessel and a 1 m^2^ Filmtec membrane SW302540 (Dow Chemical Company, Midland, TX, USA). The RO concentrate was freeze-dried in a Lyph-Lock freeze dryer (Labconco Corporation, Kansas City, MI, USA) in order to obtain the SCW powder.

### 2.2. Manufacture of the Coatings with Sheep Second Cheese Whey Powder

Whey protein isolate (WPI) (NatWPI90, 90% *w*/*w* protein, 5.5% lactose, 1.0% fat, 2.5% ashes) supplied by InLeit Ingredients S.L.U. (Curtis, Spain) and sheep’s SCW powder (54.7% *w*/*w* protein, 14.3% fat, 7.9% ashes) were weighed in order to originate a final concentration of protein in the cheese coating solution of 8% (*w*/*v*) and a protein proportion of 2:1 SCW:WPI.

The WPI was dissolved in distilled water by slow stirring for 30 min at 20 °C. Then, glycerol (Panreac, Barcelona, Spain) was added in a 1:1 protein/plasticizer ratio. The pH was adjusted to 7.0 with 0.1 M NaOH. Afterwards, the WPI:plasticizer solution was heated to 90 °C for 15 min with stirring. After cooling to 30 °C, the WPI:glycerol solution was separated into three equal portions to prepare the coatings containing sheep’s SCW as follows: coating solution without essential oils; coating solution with oregano essential oil; coating solution with clary sage essential oil.

The essential oils (oregano and clary sage; Plena Natura, Amadora Portugal) (1% of the protein weight in solution) and the SCW powder were added simultaneously to the previously prepared WPI/plasticizer solution, and the mixture was homogenized using an Ultra-Turrax (mod. T18 basic, IKA™, Staufen, Germany). The final pH was adjusted to 7 with 0.1 M NaOH, and the solution was stored in refrigeration until used.

Summarizing, the final SCW:WPI/plasticizer solution had 8% (*w*/*v*) protein (in the proportion 2:1 SCW:WPI) and a ratio of 1:1 protein/plasticizer.

### 2.3. Manufacture of Cheeses

Cheese manufacture was carried out according to Borges et al. [26] with some modifications. Cylindrical cheeses of approximately 250 g were produced at Escola Superior Agrária (ESAC, Coimbra, Portugal) dairy pilot plant and used as cheese food model for sheep’s SCW-based coating application. A local producer supplied the bovine milk (3.2% protein, 3.5% fat, 4.2% lactose). Milk pasteurization occurred at 75 °C (±0.1 °C) for 30 s in a plate and frame heat exchanger (Albinox, S. Pedro do Sul, Portugal). After stabilizing the temperature at 29.5 ± 0.5 °C, 0.04% of a 51% *w*/*v* CaCl_2_ solution (Tecnilac, Viseu, Portugal), 0.01% *w*/*v* starter culture containing *Lactococcus lactis* subsp. *lactis*, *Lactococcus lactis* subsp. *Cremoris*, and *Streptococcus thermophilus* (Mesófilo Plus Starter, Enzilab, Leça do Balio, Portugal), and granulated lysozyme (0.025% *w*/*v*) (Biostar, Rua das Picarotas, Portugal) were added. Finally, 0.0025% *w*/*v* animal rennet (>92 g/100 g chymosin, supplied by Tecnilac, Viseu, Portugal), previously diluted in water, was added. The milk coagulated in approximately 45 min, and once coagulation was complete, the curd was cut into small pieces (2 cm^3^) to promote the drainage of whey. After half of the whey was drained, an equal amount salt water (1% *w*/*v* salt, 30 °C) was added to the curd, and the mixture was thoroughly stirred before final draining of the whey for 15 min. The curds were placed in plastic molds before being pressed and stored in a refrigerated chamber at approximated 8–9 °C for 24 h. Next day, the cheeses were immersed in a brine solution (18–20 °Baumé) for 1 h 30 min. On the following day, the different coatings were applied by immersion of the cheeses on the coating solution:C: control cheese without coating;N: control cheese with natamycin;WC: cheese with SCW:WPI coating;WCO: cheese with SCW:WPI coating containing oregano essential oil;WCS: cheese with SCW:WPI coating containing clary sage essential oil.

The solution of natamycin (Nataseen™-L, Siveele, supplied by Enzilab, Leça do Balio, Portugal) containing 3.2 g/L in distilled water was previously prepared, and the cheeses were submerged in it for one minute.

The cheeses were maintained at ca. 10 °C, and the operation was repeated on the following day. Afterwards, the cheeses were transferred to the ripening room, where they were maintained for 28 days at 10 ± 2 °C, 80–90% R.H.

### 2.4. Physicochemical Analysis

Cheese sample total solids contents were measured by drying in a Schutzart DIN 40050-IP20 Memmert™ oven according to NP 1598:1983 for cheese [64]. The cheese samples’ ash content was determined through incineration after drying in a HD-23 Hobersal™ electric muffle furnace. The Gerber method was used for fat content determination in a SuperVario-N Funke Gerber™ centrifuge according to NP 2105:1983 [65]. The total nitrogen content was quantified in a Digestion System 6 1007 Digester Tecator™ using the Kjeldahl method [66]. A conversion factor of 6.38 was applied to calculate the protein content of cheeses. All determinations were performed in triplicate.

#### 2.4.1. pH and Titratable Acidity

The pH of cheeses was directly determined using a pH meter (HI 9025 HANNA Instruments) in order to monitor its changes between the 1st and the 28th days of storage. The evolution of the titratable acidity (TA) (% lactic acid) was measured by titration according to AOAC 920.124 for cheese [67].

#### 2.4.2. Water Activity

Cheese water activity (a_w_) was measured in a Rotronic Hygrolab (Bassersdorf, Switzerland) laboratory device for a_w_ measurement equipped with an HC2-AW cell at 20 °C. Water activity was determined by cutting a circle of cheese (2.25 cm radius and 0.5 cm height) and placing it in cell until a constant value of equilibrium relative humidity (E.R.H.) was attained.

#### 2.4.3. Color Parameters

The color parameters of rind and paste of cheese samples were determined (according to Borges et al. [26] with some modifications) with a colorimeter (model CR-200B Chroma Meter, Minolta™, Tokyo, Japan) using illuminant C and 10° standard observer; the aperture diameter was 1 cm. It was previously calibrated with a white standard (CR-A47: Y = 94.7; x 0.313; y 0.3204). The CIEL*a*b* system was used to measure the color coordinates. Three measurements were taken for each sample. Total color difference (ΔEab*) was calculated as follows:ΔEab* = [(L* − L*^0^)^2^ + (a* − a*^0^)^2^ + (b* − b*^0^)^2^]^1/2^(1)
where L*^0^, a*^0^, and b*^0^ and L*, a*, and b* were the values measured for the samples under comparison (C vs. N vs. WC vs. WCO vs. WCS; 1st day vs. 7th, 14th day vs. 7th day, 21th day vs. 14th day, and 28th day vs. 21st day).

#### 2.4.4. Texture Parameters

Texture analysis (TPA test) of cheese samples was carried out (according to Borges et al. [26] with some modifications) using a texture analyzer (model TA.XT Express Enhanced, Stable Micro Systems™, Godalming, UK). The penetration distance was 20 mm at 2 mm/s, and a cylindrical probe of 6 mm diameter was used. The Specific Expression PC software (version 1.1.9.0) was utilized for results calculation.

### 2.5. Microbiological Analysis

Lactic acid bacteria (LAB) of the genera *Lactobacillus* spp. and *Lactococcus* spp. were counted on M17 agar (in aerobiosis) and on MRS agar (in anaerobiosis) plates (Biokar Diagnostics, Allonne, France) at 37 °C for 48 h, respectively, according to ISO 7889 IDF 117 (2003) [68]. Yeasts and molds were enumerated according to ISO 6611 IDF 94 (2004) [69] in plates at 25 °C. Total mesophilic aerobic counts were enumerated at 30 °C for 72 h on plate count agar (PCA) according to Gonzalez-Fandos et al. [70].

Microbial analyses were carried out on the 1st, 7th, 14th, 21st, and 28th days of ripening in triplicate, together with two controls for each medium. Results are presented as the log CFU/g of cheese product.

### 2.6. Statistical Analysis

A two-way ANOVA was used to analyze the effects of days of ripening and type of cheese and their interaction on all the parameters determined. The differences among cheese samples and the effect of days of ripening were assessed using one-way ANOVA, and the means were compared using Tukey’s post hoc test.

A significance level of *p* < 0.05 was used for all mean evaluations. Statistical analyses were carried out using the IBM SPSS Statistics software for Windows version 27 (2021; IBM Corp, Armonk, NY, USA). Graphs were produced using the Statistica Software, version 8 (Stasoft Europe, Hamburg, Germany).

## 3. Results

Table 1 presents the compositional parameters of the cheese samples that, according to the Portuguese standard, are the basis for the classification of cheeses. Nonsignificant differences were observed regarding these parameters. All the cheeses produced were full-fat (≥45 <60% fat in dry matter) and semihard cheeses (54–63% moisture in the defatted cheese).

The evolution of the dry matter and of the water activity of the different cheeses over ripening can be observed in Figure 1. Significant differences regarding dry matter and the water activity were observed among samples and during ripening, as can be observed in Table 2. The variation in dry matter was most evident between the 1st and the 7th day. The dry matter of the WC cheese showed a very significant increase between the 14th and the 21st day and presented the highest content of solids by the end of storage (Appendix A). The values of dry matter at the end of ripening were lower in cheeses containing the coating with essential oils. This fact was reflected in the significantly higher values of water activity of these samples (Appendix A).

Significant differences were also observed for the pH and the titratable acidity (TA) of the different cheeses (Figure 2). Control cheeses showed significantly higher values of TA than the other cheeses at the first day of ripening; these cheeses also presented higher values of TA at the other days of ripening except in the 21st day. It was observed that, despite its lower pH, the cheese with oregano essential oil presented the lowest titratable acidity by the end of the ripening time. Lower values of TA were also observed for WC and WCS samples compared to C and N cheeses at the end of ripening.

Regarding the color parameters, the reduction in lightness of the rind over storage (Figure 3A) and the increase in parameter b* (increasing yellowness) from the first to the 7th day (Figure 3E) is clear. Parameter b* of the rind increased from values around 15 to values of ca. 20 between the 1st and the 7th day of ripening and were maintained, or slightly reduced, at the end of ripening. Parameter b* of the paste (Figure 3F) had the same pattern as was observed in the rind, but, in this case, the b* values increased to just 16–19.

The lightness of the paste of all samples (Figure 3B) also decreased, but the decrease was less pronounced when compared to the color of the rind. Parameter a* (green-red axis) of the rind and of the paste of cheeses (Figure 3C,D) decreased from the first to the 7th day, thus showing an increase toward the end of ripening, which was less pronounced on the paste. The exception was the paste of the control cheese, in which a* decreased with storage time.

Table 3 presents the ΔEab* values between cheese samples (compared to the control—C) and for the same sample at different days of storage (compared to the 1st day of storage). It is evident that the differences between the cheese samples at the same day of storage are less marked than the differences observed for the same sample at different days of storage. At the first day of storage, the ΔEab* values between the samples and the control were lower than one, thereby indicating that a common observer cannot detect color differences. Exceptions were WC vs. C and WCS vs. C. The differences between samples tended to increase over ripening and reached the highest values observed at the 21st day. On the 28th day, the color differences were not so marked as at the 21st day. On the contrary, considering the evolution of the color of each cheese sample over time, very high ΔEab* values were observed between the 1st and the 7th day. Lower values were observed between the 14th and the 7th day, thereby increasing between the 21st and the 14th day. Between the 28th and the 21st day, the color differences were less marked, both in the rind and in the paste.

Taking into attention the values of the two-way ANOVA test (Table 4), significant differences among products and days of ripening regarding the color parameter of the rind and of the paste were detected. However, as it can be seen in Appendix A, no significant differences were observed among different cheeses in all the color parameters (L*, a*, and b* of the rind and of the paste) at the end of the ripening.

Figure 4 presents the external aspect of the representative cheeses over the storage period. It can be observed that all cheeses showed low external contamination with molds over storage. However, at the 21st and 28th days, some cheeses containing oregano EO presented some visible contamination by molds.

Figure 5 displays the texture parameters of the cheese samples. Significant differences were observed for all the parameters as a result of the ripening time (Table 5). The cheeses with higher ripening times showed significantly higher values for hardness, chewiness, and gumminess, as well as significantly lower values for adhesiveness and resilience (Appendix A). The cheeses containing SCW coatings presented lower values for hardness, chewiness, gumminess, and cohesiveness (Figure 5A,C–E) when compared to C and N.

The microbiological evaluation of the different cheese samples is displayed in Figure 6. Significant differences were observed for all microbial groups over storage (Table 6). In all cases, the counts of lactobacilli and lactococci surpassed Log 7–8 CFU/g, while the counts of yeasts and molds increased steadily from ca. Log 3 CFU/g to Log 6 CFU/g. The type of cheese coating significantly influenced the counts of lactobacilli, yeasts, and molds (Table 6). The lowest counts of yeasts and molds were observed in the samples with natamycin. However, nonsignificant differences were observed between products at the end of storage (Appendix A).

## 4. Discussion

Dairy products, namely cheeses, are commonly contaminated with molds, which produce visible or non-visible defects that lead to significant economic losses. Methods to reduce contamination include good manufacturing and hygiene practices, as well as air filtration and the manual cleaning of cheeses during ripening. Control methods such as inactivation treatments, temperature control, or modified atmosphere packaging are also used. However, fungal spoilage remains a problem for cheese producers despite technological advances in existing preservation methods. New preservation technologies have been introduced in recent years, including bioprotective cultures or edible coatings containing antimicrobial agents [62,63,71,72,73,74,75].

We only found one work referring the use of SCW as a material to produce edible coatings. In the work performed by Alfano and coworkers [58], SCW was microfiltered, and the MF permeate was concentrated by UF using 100 kDa membranes. The UF permeate obtained was also concentrated using 10 kDa membranes. The 10 kDa permeate was further concentrated on nanofiltration membranes. With this approach, different retentate fractions were obtained and used to make films or to serve as a primary substrate for probiotic cultures. These authors report that only extensively diafiltrated SCW powder, with more than 50% protein on a dry weight basis, proved to form films in the presence of glycerol.

In the present work, SCW was immediately concentrated by UF (10 kDa), and the UF concentrate was diafiltrated to reduce lactose and to increase the ratio of nitrogen components on the SCW powder. Hence, a powder with almost 55% (*w*/*w*) crude protein was obtained. Our preliminary essays aiming at evaluating the potential of sheep’s SCW diafiltrated powder as a potential material to produce edible films and coatings indicated that the SCW powder showed poor film forming properties. The best film forming properties were obtained by mixing SCW powder with WPI in the proportions 2:1 or 1:1. The amount of fat in our SCW and the strong thermal treatment used for the production of whey cheeses, as is the case of *Requeijão*, might have affected the film forming properties of the sheep’s SCW powder. Therefore, the decision regarding the proportions of both types of whey proteins was based on the maximization of the utilization of SCW powder (i.e., 2:1 SCW:WPI).

Regarding the physicochemical characteristics of the cheese samples, it is worth noting the beneficial effect that oregano and clary sage EOs added to SCW:WPI coatings with respect to the retention of moisture. Water loss reduction in cheeses with coatings containing oregano essential oil has also been reported by Artiga-Artigas et al. [76]. This is a consequence of the vapor permeability reduction due to the decrease in films’ hydrophilic portion due to the incorporation of essential oil, as reported by Lee and coworkers [77]. The addition of a hydrophobic agent to hydrophilic polymer matrices decreases their affinity to the water and enhances their ability to act as a barrier to water vapor permeation [78]. This fact had a particular influence on the texture properties of samples WCO and WCS, with the reduction of hardness, chewiness, and gumminess, which can be considered positive regarding the desired sensory characteristics of the cheese type tested. Cheeses with whey protein coatings (with or without essential oils) exhibited lower hardness values than control batches at 28 days of storage. Decreases in this parameter have also been observed in cheeses coated with hydrolyzed WPC solutions with added marjoram essential oil [79].

Despite the significant differences regarding the microbiological counts of the different cheese samples obtained, it can be considered that the antimicrobial agents used in the coatings (natamycin, oregano essential oil, and clary sage essential oil) did not affect the bacteria of the starter culture used. In all cases, the lactobacilli and lactococci counts surpassed log 8.0 CFU/g. Exceptions were the lactobacilli counts of the N and WCS samples at the day after production (log 7.6 and 7.8 CFU/g respectively). EOs may negatively affect the growth of lactic acid bacteria of cheese starter cultures, but they are relatively resistant to their antimicrobial compounds. It has also been reported that they are not affected [78]. In the present study, lactobacilli counts were not significantly affected by EO addition to SCW, while lactococci counts increased in SCW-coated cheeses.

In all cases, the counts of yeasts and molds were lower than log 4.5 CFU/g until the 7th day, thereby increasing steadily to ca. log 6 CFU/g at the end of ripening. The lowest value (5.6 log CFU/g) at the end of ripening was observed for the sample containing natamycin. Hence, it cannot be concluded that either the commercial coating or the coatings based on SCW with EOs were more efficient in reducing yeast and mold counts when compared to the negative control (C).

The antimicrobial activity of the coatings with essential oils depends on the EO concentration and also on the active compounds; in oregano EO is mainly carvacrol, and in sage EO are camphor, α-thujone, and 1,8-cineole [79,80]. However, the efficacy of essential oils depends on various factors, including the microorganism strains, the EO composition and concentration, the environmental conditions, and the chemical composition of the food to which they are added. If the EO is incorporated into a film or coating, the composition of the latter may also modify its effects. In dairy products, proteins could interact with phenolic compounds and limit the EO activity against microorganisms [78]. It has been reported in sodium alginate-based coatings [76] that concentrations of oregano EO lower than 2% *w*/*w* seem not to be enough to produce the inhibition of pathogenic and spoilage microorganisms (e.g., molds and yeast) in cheese.

Several authors report on the positive effect of Eos as antimicrobial agents in whey films as such or applied as coatings in cheese. Fahrullah and coworkers [36] report that the addition of clove essential oil at different levels inhibited the growth of *E. coli*, *Salmonella* sp., *S. aureus*, and molds in whey-based films. However, high incorporation levels were used (5–15%), and this fact may influence the sensory properties of the food products in which the films are applied.

Çakmak and coworkers [37] tested lemon and bergamot essential oils as antimicrobials in whey-based films. Although bergamot oil had a strong antimicrobial activity against *E. coli* and *S. aureus*, lemon oil was a poorer antimicrobial against *S. aureus*. Both EOs also showed efficacy against *Aspergillus niger*. These authors also refer that water vapor transmission rates and water vapor permeability of edible films are important factors to be considered, and it is expected that both the plasticizer and the EO composition affect those parameters. Their results demonstrated that 3.5% EO in the film composition originated a better barrier property against water vapor. This was also observed in our work, even with lower levels of EOs addition.

Seydim and coworkers [38] tested WPI films containing 2% (*w*/*v*) oregano and garlic EOs, nisin, and natamycin. *E. coli* O157:H7 inoculated in Kasar cheese containing oregano EO showed significant decreases on the first day and at the end of storage in comparison to cheeses without WPI film. Similar results were obtained with nisin. No significant *Penicillium* spp. reduction was observed on the cheese samples with antimicrobial WPI film on the first day. The cheese samples with natamycin presented the highest microbial reduction. At day 15, the cheese samples with oregano EO showed a 1.20 log reduction of *Penicillium* spp. compared to the control groups without WPI film. Similarly, the cheese with natamycin showed a 1.78 log reduction in comparison to the control. The WPI films incorporated with oregano EO exhibited higher antimicrobial activity against *E. coli* O157:H7, while the higher bactericidal effect on *L. monocytogenes* was observed in WPI films containing nisin. Cheese samples containing garlic EO showed higher microbial content when compared with cheeses containing oregano EO. Cheeses containing natamycin presented lower levels of *Penicillium* spp. In the present study, natamycin also showed the best results regarding the inhibition of molds.

Seydim and Sarikus [39] refer that the direct addition of EOs to foods decreases bacterial populations but may change the sensory characteristics of the products. Oregano and garlic Eos exhibited greater inhibitory zones on *S. aureus*, *S. enteritidis*, *L. monocytogenes*, *E. coli*, and *L. plantarum* than rosemary EO-incorporated films. Ture et al. [81] investigated the antifungal effect of films containing natamycin on Kasar cheeses inoculated with *Aspergillus niger* and *P. roqueforti* during 30 days of storage. The authors found that protein-based films were more effective than carbohydrate-based films. An oregano EO antifungal effect was also observed on chitosan-based coatings on cheeses [82].

Bahram and coworkers [41] report that films containing cinnamon (CEO) (0.8 and 1.5% *v*/*v*) exhibited remarkable antibacterial activity against both Gram-positive and Gram-negative strains, as well as a good inhibitory effect on the fungi tested. However, no inhibition zone was observed in the film disk at a concentration of 0.8% CEO. With 1.5%, a clear inhibition zone was observed.

Ramos and coworkers observed that the addition of natamycin to edible films manufactured with whey protein isolate and glycerol had an important effect against yeasts but had no activity against bacteria [47]. Fajardo and coworkers observed that the addition of natamycin to coatings with chitosan caused a reduction in molds and yeasts in cheese samples after 27 days of storage [83]. Yangilar [46] reported that the application of films only with chitosan and whey proteins to cheeses reduced the mold counts compared to the cheeses without these films.

As a final remark, taking in consideration that, in Portugal, most of the SCW produced results from ovine whey cheeses, which are curdled with vegetable rennet (*Cynara cardunculus*), and since this curdling agent has an intense proteolytic activity that can generate peptides with biological impact, namely antihypertensive and antimicrobial activities, the use of ovine SCW-based edible coatings resulting from such cheeses can ultimately reveal such biological functions. In addition, the previous fermentation of SCW by lactic acid or probiotic bacteria can further enhance the potential for the recovery and valorization of such a by-product. Therefore, it is important to develop research activities aiming at highlighting the potential of SCW as a potential ingredient of functional food products.

## 5. Conclusions

Second cheese whey, the by-product of whey cheese production, due to its composition, which includes valuable proteins and other nutrients, is an object of food technologists’ interest. The study demonstrated that SCW resulting from ovine whey cheese production may be used to prepare edible cheese coatings, which additionally were fortified with oregano and clary sage essential oils. The coatings were used to protect the surface of a bovine milk rennet curdled cheese model. The cheese with SCW coatings without and with EOs were characterized by lower titratable acidity in comparison to the control samples. The results also indicate that WCS and WCO coatings prevent the cheeses from excessive evaporation, which results in higher moisture. Moreover, the samples containing SCW coatings had lower values of hardness, chewiness, and gumminess at the end of ripening. The differences in color between samples with and without coatings were not evident. Coatings based on SCW did not affect the viability of the starter bacteria cultures. However, tested films were not more efficient in reducing yeast and mold counts comparing to the control cheese without coating and to cheese with a commercial coating containing natamycin. To conclude, SCW:WPI coatings without and with oregano and clary sage EO may successfully substitute commercial coatings.

## Figures and Tables

**Figure 1 foods-13-00674-f001:**
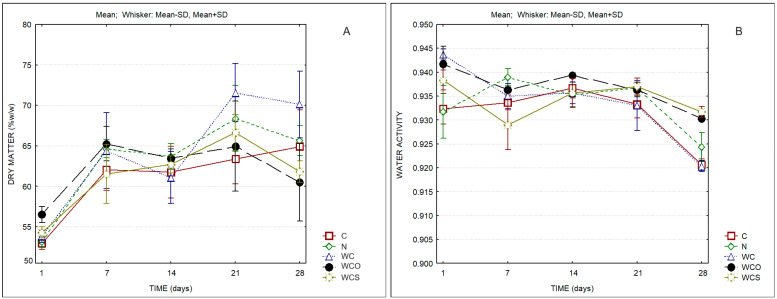
Dry matter (**A**) and water activity (a_w_) (**B**) of cheese samples. C—Control; N—Cheese with natamycin coating; WC—Cheese with SCW coating; WCO—Cheese with SCW coating with oregano essential oil; WCS—Cheese with SCW coating with clary sage essential oil.

**Figure 2 foods-13-00674-f002:**
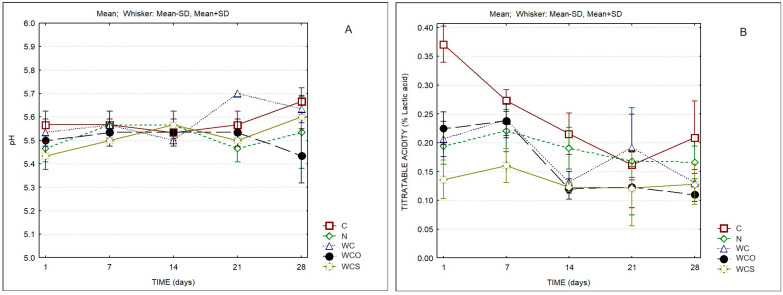
pH (**A**) and titratable acidity (**B**) of cheese samples. C—Control; N—Cheese with natamycin coating; WC—Cheese with SCW coating; WCO—Cheese with SCW coating with oregano essential oil; WCS—Cheese with SCW coating with clary sage essential oil.

**Figure 3 foods-13-00674-f003:**
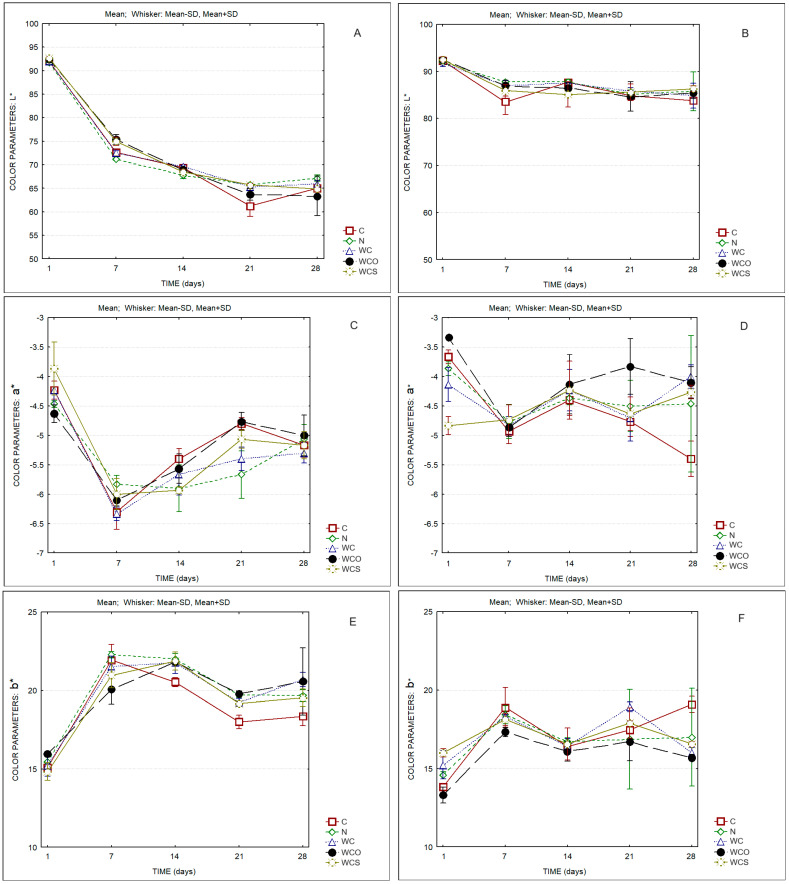
Color parameters of cheese samples. (**A**) L* rind; (**B**) L* paste; (**C**) a* rind; (**D**) a* paste; (**E**) b* rind; (**F**) b* paste. C—Control; N—Cheese with natamycin coating; WC—Cheese with SCW coating; WCO—Cheese with SCW coating with oregano essential oil; WCS—Cheese with SCW coating with clary sage essential oil.

**Figure 4 foods-13-00674-f004:**
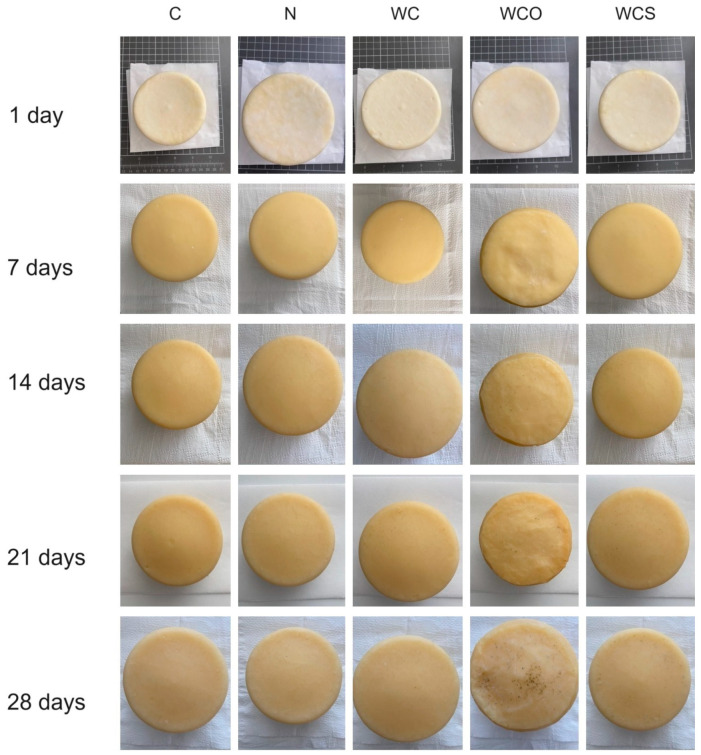
External aspect of representative cheese samples over the ripening period. C—Control; N—Cheese with natamycin coating; WC—Cheese with SCW coating; WCO—Cheese with SCW coating with oregano essential oil; WCS—Cheese with SCW coating with clary sage essential oil.

**Figure 5 foods-13-00674-f005:**
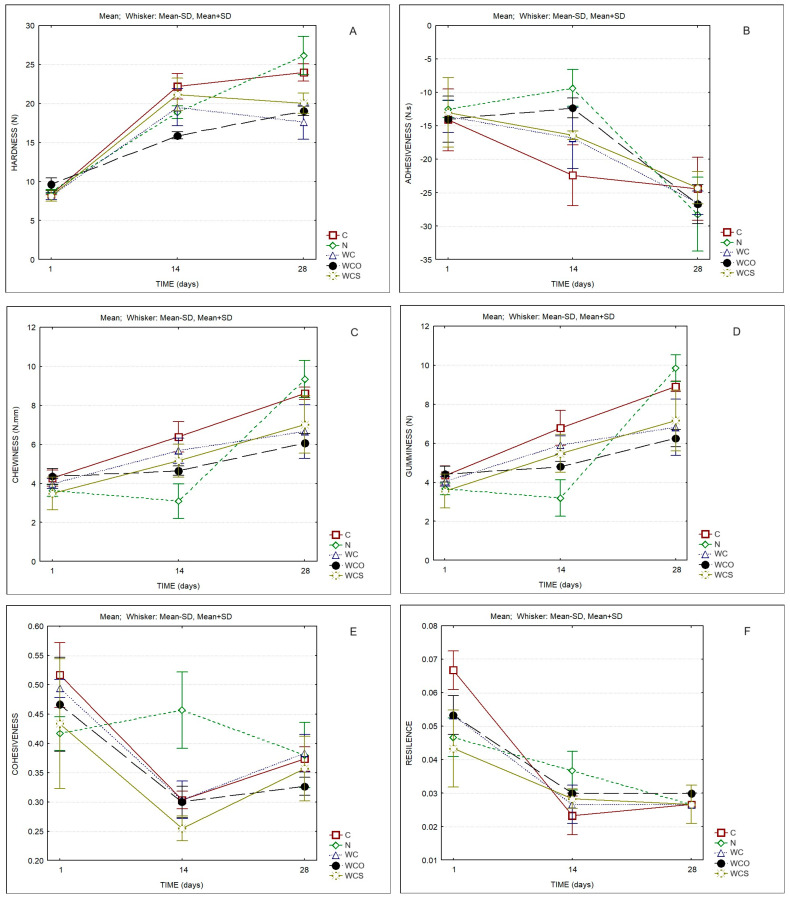
Texture parameters of cheese samples. (**A**) Hardness; (**B**) Adhesiveness; (**C**) Chewiness; (**D**) Gumminess; (**E**) Cohesiveness; (**F**) Resilience. C—Control; N—Cheese with natamycin coating; WC—Cheese with SCW coating; WCO—Cheese with SCW coating with oregano essential oil; WCS—Cheese with SCW coating with clary sage essential oil.

**Figure 6 foods-13-00674-f006:**
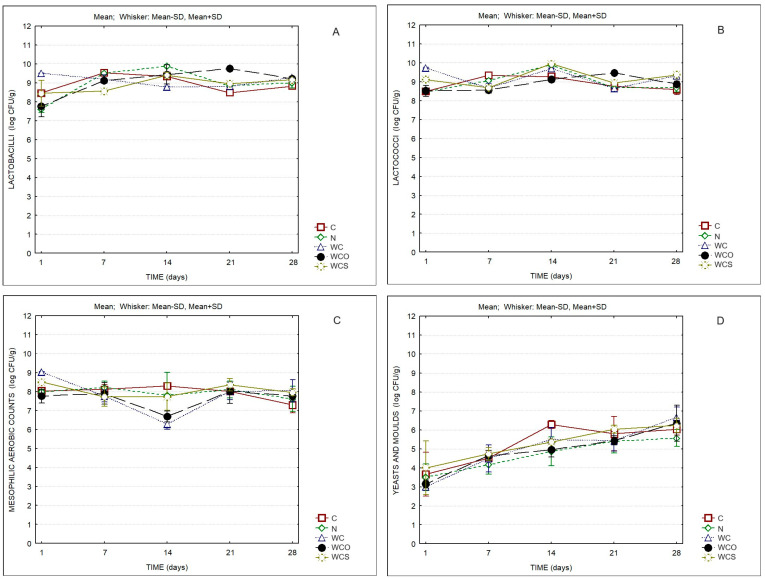
Microbiological parameters of cheese samples. (**A**) Lactobacilli; (**B**) Lactococci; (**C**) Mesophilic aerobic counts; (**D**) Yeasts and molds. C—Control; N—Cheese with natamycin coating; WC—Cheese with SCW coating; WCO—Cheese with SCW coating with oregano essential oil; WCS—Cheese with SCW coating with clary sage essential oil.

**Table 1 foods-13-00674-t001:** Gross chemical composition of cheeses.

Cheese Sample	Protein in Dry Matter (%*w*/*w*)	Fat in Dry Matter (%*w*/*w*)	Moisture in Defatted Cheese (%*w*/*w*)
C	31.05 ± 1.08 ^a^	46.02 ± 1.43 ^a^	61.27 ± 1.40 ^a^
N	28.92 ± 1.30 ^a^	48.00 ± 2.16 ^a^	62.86 ± 0.91 ^a^
WC	30.08 ± 1.17 ^a^	48.07 ± 1.79 ^a^	62.28 ± 1.26 ^a^
WCO	31.89 ± 3.06 ^a^	49.26 ± 0.89 ^a^	60.73 ± 0.78 ^a^
WCS	29.61 ± 0.85 ^a^	49.07 ± 2.06 ^a^	62.79 ±0.57 ^a^

C—Control; N—cheese with natamycin; WC—cheese with SCW coating; WCO—cheese with SCW coating with oregano essential oil; WCS—cheese with SCW coating with clary sage essential oil. Same superscript letter (a) in a column indicates no significant differences.

**Table 2 foods-13-00674-t002:** Two-way ANOVA of physicochemical characteristics of cheese samples.

Physicochemical Analysis		Two-Way ANOVA	
		*F*	*p* Value
Dry matter	Time	41.42	0.000
	Product	2.92	0.030
	Interaction	1.89	0.045
a_w_	Time	48.83	0.000
	Product	7.90	0.000
	Interaction	5.98	0.000
pH	Time	2.91	0.031
	Product	5.68	0.000
	Interaction	2.82	0.003
Titratable acidity	Time	16.68	0.000
	Product	17.35	0.000
	Interaction	2.67	0.004

**Table 3 foods-13-00674-t003:** ΔEab* values of cheese samples over storage.

ΔEab* values between products at different ripening days					
Ripening time (days)	1	7	14	21	28
Cheese samples			RIND		
N vs. C	0.4 ± 0.7	1.6 ± 0.8	2.7 ± 1.1	17.9 ± 10.3	4.2 ± 4.9
WC vs. C	0.7 ± 0.5	0.3 ± 0.4	1.2 ± 0.9	19.2 ± 13.0	5.4 ± 5.3
WCO vs. C	0.6 ± 0.2	7.1 ± 6.5	1.0 ± 0.5	8.5 ± 10.2	15.0 ± 12.2
WCS vs. C	0.7 ± 0.7	3.5 ± 2.0	1.7 ± 1.4	12.7 ± 11.3	2.4 ± 2.4
			PASTE		
N vs. C	0.3 ± 0.1	12.2 ± 11.7	0.4 ± 0.4	3.9 ± 4.7	14.6 ± 9.3
WC vs. C	1.7 ± 1.1	8.0 ± 8.5	0.2 ± 0.2	2.2 ± 2.9	9.3 ± 3.6
WCO vs. C	0.4 ± 0.4	11.4 ± 16.6	1.8 ± 2.7	6.5 ± 4.7	8.9 ± 0.6
WCS vs. C	3.2 ± 0.6	4.5 ± 4.4	5.6 ± 7.9	1.0 ± 1.5	7.4 ± 1.4
ΔEab* values for the same product at different ripening days					
Cheese Sample	C	N	WC	WCO	WCS
Ripening time (days)			RIND		
7 vs. 1	213.5 ± 15.6	234.5 ± 9.2	210.0 ± 21.1	155.3 ± 23.4	179.4 ± 26.1
14 vs. 7	7.5 ± 0.8	6.3 ± 3.0	4.9 ± 1.5	22.4 ± 5.7	21.6 ± 4.8
21 vs. 14	39.7 ± 19.1	16.2 ± 20.1	37.1 ± 24.9	16.7 ± 4.7	8.2 ± 2.5
28 vs. 21	7.7 ± 2.5	7.2 ± 9.5	18.9 ± 15.2	9.7 ± 7.4	0.6 ± 0.2
			PASTE		
7 vs. 1	55.4 ± 32.0	16.9 ± 3.4	17.2 ± 4.5	26.0 ± 11.4	24.3 ± 5.3
14 vs. 7	14.3 ± 13.7	1.8 ± 0.9	2.7 ± 0.8	2.4 ± 1.3	3.0 ± 2.4
21 vs. 14	5.2 ± 2.6	7.3 ± 3.5	4.8 ± 1.6	3.7 ± 2.6	2.5 ± 1.9
28 vs. 21	2.8 ± 1.2	9.3 ± 2.3	8.6 ± 2.2	5.6 ± 5.1	2.2 ± 0.2

**Table 4 foods-13-00674-t004:** Two-way ANOVA of color parameters of cheese samples.

Color Parameters		Two-Way ANOVA	
Rind		*F*	*p* Value
L*	Time	1478.07	0.000
	Product	2.71	0.040
	Interaction	4.74	0.000
a*	Time	114.13	0.000
	Product	2.56	0.050
	Interaction	3.49	0.000
b*	Time	100.44	0.000
	Product	3.07	0.025
	Interaction	3.63	0.000
Paste			
L*	Time	51.68	0.000
	Product	1.35	0.264
	Interaction	1.31	0.227
a*	Time	11.06	0.000
	Product	5.44	0.001
	Interaction	3.09	0.001
b*	Time	25.26	0.000
	Product	3.73	0.010
	Interaction	1.80	0.059

**Table 5 foods-13-00674-t005:** Two-way ANOVA of texture parameters of cheese samples.

Texture Parameters		Two-Way ANOVA	
		*F*	*p* Value
Hardness	Time	267.79	0.000
	Product	9.94	0.000
	Interaction	25.80	0.000
Adhesiveness	Time	47.96	0.000
	Product	1.28	0.300
	Interaction	2.45	0.038
Chewiness	Time	84.76	0.000
	Product	4.40	0.007
	Interaction	6.72	0.000
Gumminess	Time	90.62	0.000
	Product	4.98	0.004
	Interaction	7.72	0.000
Cohesiveness	Time	30.28	0.000
	Product	2.54	0.061
	Interaction	3.40	0.007
Resilience	Time	88.87	0.000
	Product	1.46	0.239
	Interaction	3.99	0.003

**Table 6 foods-13-00674-t006:** Two-way ANOVA of microbiological characteristics of cheese samples.

Microbiological Analysis		Two-Way ANOVA	
		*F*	*p* Value
Lactococci	Time	35.66	0.000
	Product	9.58	0.000
	Interaction	12.63	0.000
Lactobacilli	Time	22.64	0.000
	Product	1.17	0.343
	Interaction	8.65	0.000
Mesophilic Aerobic Bacteria	Time	9.37	0.000
	Product	2.27	0.070
	Interaction	3.48	0.000
Yeasts and Molds	Time	52.60	0.000
	Product	3.13	0.019
	Interaction	1.29	0.225

## Data Availability

All data are available in the manuscript and Appendix A. Further inquiries can be directed to the corresponding author.

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
