# Peer review of "Sheep’s Second Cheese Whey Edible Coatings with Oregano and Clary Sage Essential Oils Used as Sustainable Packaging Material in Cheese"

_foods, 2024, doi:10.3390/foods13050674_

Round 1
Reviewer 1 Report
Comments and Suggestions for Authors
This manuscript described the application of whey edible coatings with oregano and clary sage essential oils in cheese. At first glance, the topic is interesting and brings one more application of innovative technology. However, the authors should clearly state the main novel aspects of this approach against previous research. The major remarks are listed below:
-Line 19-24 should be summarized. Research purpose, the novelty or importance of your work, and the main results should be highlighted.
- Please add relevant previous publications regarding the coating of cheese in the introduction section.
-Sections 2.2 and 2.3: Relevant references should be added.
-Line 137: This concentration of natamycin (3200 ppm) is safe in food? Support your answer via references. How much was the amount of natamycin in the cheese sample after coating?
- The order of mentioned references in the text is incorrect.
-Line 147: What is the meaning of N content?
-2.4.2, 2.4.3. and 2.4.4. References should be added.
-Figures: The quality of figures should be improved.
- Table 3, STD, and the result of statistical analysis should be added.
- Please replace updated references instead of old ones.
Comments on the Quality of English Language
This manuscript described the application of whey edible coatings with oregano and clary sage essential oils in cheese. At first glance, the topic is interesting and brings one more application of innovative technology. However, the authors should clearly state the main novel aspects of this approach against previous research. The major remarks are listed below:
-Line 19-24 should be summarized. Research purpose, the novelty or importance of your work, and the main results should be highlighted.
- Please add relevant previous publications regarding the coating of cheese in the introduction section.
-Sections 2.2 and 2.3: Relevant references should be added.
-Line 137: This concentration of natamycin (3200 ppm) is safe in food? Support your answer via references. How much was the amount of natamycin in the cheese sample after coating?
- The order of mentioned references in the text is incorrect.
-Line 147: What is the meaning of N content?
-2.4.2, 2.4.3. and 2.4.4. References should be added.
-Figures: The quality of figures should be improved.
- Table 3, STD, and the result of statistical analysis should be added.
- Please replace updated references instead of old ones.
Author Response
We thank you for your helpful suggestions and corrected the manuscript as indicated.

Reviewer 2 Report
Comments and Suggestions for Authors
In the current study, Arona Pires prepared several edible coatings for bovine milk cheeses using the by-product resulting from whey cheese production. The idea is novel and it has great industrial application potential. The manuscript is well-written and the analysis was properly performed.
I only have several minor questions/ suggestions the authors can consider:
1) Is there any evaluation about the cost of this SCW edible coating and the commercial coating containing natamycin?
2) In figure 2 B, why the samples without coating showed obvious higher titratable acidity at the beginning of the experiments (D1)? The authors only discussed the situations in the end of ripening.
3) The authors determined the influence of different coatings on the microbial groups in cheese over storage. Is there any regulations or threshold for the presence of these bacteria / yeast/ moulds. Consequently, could the observed changes in microbial groups upon coating also influence the shelf life of the cheese.
Author Response
Dear reviewer. We thank you for your helpful suggestions and corrected the manuscript as indicated.
